# Knowledge, Attitudes, and Practices in Antibiotic Dispensing amongst Pharmacists in Trinidad and Tobago: Exploring a Novel Dichotomy of Antibiotic Laws

**DOI:** 10.3390/antibiotics12071094

**Published:** 2023-06-23

**Authors:** Rajeev P. Nagassar, Amanda Carrington, Darren K. Dookeeram, Keston Daniel, Roma J. Bridgelal-Nagassar

**Affiliations:** 1Department of Microbiology, The Sangre Grande Hospital, The Eastern Regional Health Authority, Sangre Grande, Trinidad and Tobago; 2Department of Health Sciences, The University of Trinidad and Tobago, Trinidad and Tobago; 3Department of Emergency Medicine, The Sangre Grande Hospital, The Eastern Regional Health Authority, Sangre Grande, Trinidad and Tobago; 4The Public Health Observatory, The Eastern Regional Health Authority, Sangre Grande, Trinidad and Tobago; 5Manager, Medical Research and Audit, Directorate of Women’s Health, The Ministry of Health, Trinidad and Tobago; bridgerom@yahoo.com

**Keywords:** antibiotics, legislations, laws, dispensing, pharmacists

## Abstract

The inappropriate consumption, use, and dispensing of antibiotics are problems faced globally, with a pattern of inappropriate consumption differing in higher-income countries due to the ease of accessibility of antibiotics. The main drivers of consumption and inappropriate use are the over-the-counter sales of antibiotics by pharmacies. Trinidad and Tobago (T&T), a twin island state in the Caribbean, has two Acts of Parliament that regulate antibiotics: the Antibiotics Act and the Food and Drug Act, yet the Over-the-Counter (OTC) sale of antibiotics still exists. This study sought to determine the knowledge, attitudes, and practices regarding the OTC dispensing of antibiotics in T&T. A cross-sectional study gathered data from pharmacists in both the private and public sectors of Trinidad over 7 months. The results showed that antibiotic resistance and antibiotic abuse were seen as significant problems. The level of experience, gender (female), and age (younger) were significantly associated with having good overall knowledge of good dispensing habits and antibiotic laws (*p* = 0.036, *p* = 0.047, and *p* = 0.001, respectively). Pharmacists generally agreed that antibiotics under the Food and Drug Act may have contributed to OTC dispensing in the private sector (*p* = 0.013) and that all antibiotics should be under the Antibiotic Act (*p* = 0.002). Additionally, it was found that the dispensing of antibiotics OTC in the private sector (*p* = 0.006) occurred: without doctors’ advice and without requesting prescriptions; because it was perceived as lawful (especially by older pharmacists); and because of the perceived motivation of profit. Regulation enforcement was perceived as deficient. OTC dispensing for reasons, such as misunderstanding of laws, occurs in T&T.

## 1. Introduction

The inappropriate use and dispensing of antibiotics are global problems, with antibiotics being one of the most commonly sold drugs worldwide [1,2]. Dache et al. found in their 2021 study that “pharmacies (57%) and family members or neighbors were common sources of antibiotics in low-income countries”. Browne et al. (2021), however, found that higher levels of antibiotic consumption occurred in high-income countries, such as the United States of America, and lower consumption levels in lower-income countries, such as those in Sub-Saharan Africa, where they may be less accessible [3]. 

The inappropriate use of antibiotics has been seen as a complex problem involving various actors from the human, environmental, food, and veterinary sectors and fueled by the ‘over-the-counter’ sale of medication, dispensing by inappropriate pharmacy staff, and the use of leftover or borrowed antibiotics [2,4,5].

The most significant impact of dispensing, if not properly regulated, is the likelihood of people developing antibiotic resistance. In many cases, conflicting interests, such as profits for the prescriber and dispenser, are prioritized over preventing antimicrobial resistance and drive inappropriate dispensing behaviors [4,5,6,7,8,9,10]. These patterns will ultimately result in increased drug use and patient treatment costs in the future.

In Trinidad and Tobago, a high-income, developing country, the pharmacy is an important point of contact for patients, especially within the community. Legislative provisions in the country establish the powers and responsibilities of drug sales and distribution. This legal framework includes the “Food and Drugs Act (Act 8 of 1960), the Antibiotics Act (Act 18 of 1948), the Dangerous Drugs Act (Act 38 of 1991), the Narcotic Control (General Provisions) Regulations, the Narcotic Control (Licensing) Regulations, and the Pharmacy Board Act” (Act 7 of 1960) [6,7,8,9,10,11,12,13,14].

Antibiotics, however, are primarily regulated by the Drug Inspectorate Division (DID); this division plays a major role in private and public pharmacy inspections across the country. The Chemistry, Food, and Drug Division, another regulation agency with drug inspectors who have some operating power concerning the regulation of antibiotics, can be considered to operate within a “grey area” in the legislation. Both divisions operate under the Ministry of Health of Trinidad and Tobago but are considered to have separate roles and staff to conduct their monitoring and regulation activities. Even with the existence of two independent regulating bodies, the Trinidad and Tobago Pharmaceutical Country Profile states that antibiotics are often sold over the counter without the use of a prescription [6,7]. 

The WHO Policy Guidance on Integrated Antimicrobial Stewardship Activities lists the five (5) pillars of good antimicrobial stewardship as Commitment, Prevention, Detection, Optimization of Use, and Surveillance [8]. Pillars 2 and 3 (Prevention and Detection) will be mentioned in this study for the specific purpose of reviewing Antimicrobial Stewardship (AMS). These pillars are key to strengthening the Global Action Plan (GAP) to target the appropriate use of antimicrobials and thus strengthen antimicrobial stewardship (AMS) [8,9,10]. 

It is against this background that this Knowledge, Attitudes, and Practices (KAP) study aims to gather baseline data on the knowledge, attitudes, and practices of over-the-counter dispensing of antibiotics by pharmacists under a dichotomy of legislation in Trinidad and Tobago. It should be noted, however, that this study was conducted during the COVID-19 pandemic, with several restrictions preventing normal data collection. It also dealt with the sensitive issue of possible infringement of the law.

## 2. Results

A total of 104 responses were received from the public and private sectors. The response rate for the survey was (104/145) 71.7%, or approximately 72%. The majority (49%) of the respondents were between 21 and 40 years old, were female, had greater than 4 years of pharmacy dispensing experience, and worked in the private sector. Most respondents had four or more years of experience. Respondents were from the public, private, and both sectors.

### 2.1. Knowledge

Significant relationships were underlined. N/A means not applicable and is utilized as the response was 100% for a particular field, making this significant. The response to knowledge-related questions was yes or no and displayed as numbers and percentages, n (%). The responses are displayed in Table 1.

All respondents indicated that antibiotic resistance is a serious public health problem facing the world, regardless of age, gender, or the sector they were employed in. 

#### 2.1.1. Age

A Chi^2^ test of independence was performed to examine the relationship between age and repeated use of the same antibiotic in terms of resistance (*p* = 0.027). Additionally, Fisher’s Exact *t*-test showed that younger pharmacists displayed a significant difference in their knowledge of the fact that superbugs, such as MRSA and carbapenem-resistant Gram-negative bacilli, result in an increased length of stay for patients on wards (*p* = 0.001). 

#### 2.1.2. Sector

When pharmacists’ responses to the statement “antimicrobial resistance results in resistance to antibiotics only” were compared with their sector of employment, a significant association was found for both the private and public sectors (*p* = 0.017). This was also true for the statement “vaccines can prevent unnecessary antibiotic use and thus antibiotic resistance” and their sector of work, where more private sector pharmacists responded affirmatively (*p* = 0.001). This is displayed in Table 1. 

With regards to knowledge of the Antibiotic Act and Pharmacy Board Act, yes/no responses were gathered from the responding pharmacists and displayed as both numbers and percentages, n (%). No significant relationship was found between age, sex, experience, or sector and any question asked in this section. The Chi^2^ value for the fields “The Pharmacy Board Act regulates the dispensing of antibiotics by pharmacists” and “Have you heard of the Antibiotic Inspectorate/Drug Inspectorate?” was used. Fisher’s Exact *t*-test was used for the field “Have you heard of the Antibiotic Act?” The *p* values were insignificant, but over 80% of pharmacists in this study responded affirmatively to these fields, as displayed in Table 2.

With regards to knowledge of the Food and Drugs Act and Antibiotic Act (This is displayed in Table 3), questions were again answered yes or no, and the numbers and percentages displayed n (%). 

#### 2.1.3. Age and Experience 

Are Antibiotics registered under the Food and Drug Act under the purview of the Antibiotic Inspectorate/Drug Inspectorate? Notably, for this field with regards to age, there was no significant association for age (*p* = 0.05). With regards to the relationship between the responses to the question “Are antibiotics registered under the Food and Drug Act under the purview of the Antibiotic /Drug Inspectorate?” and years of experience, there was a significant relationship (*p* = 0.036). This is seen in Table 3. 

#### 2.1.4. Sex

When “Are any antibiotics registered under the Food and Drug Act?” was compared to the sex of the respondents, a significant relationship was noted (*p* = 0.047). Overall, 80.8% of pharmacists responded with the correct answer to this question (see Table 3).

The Chi^2^ value and the *p* values were insignificant for most other values regarding knowledge of the various Acts of Parliament. The antibiotics named were ciprofloxacin, co-trimoxazole, o-amoxiclav, cefuroxime, azithromycin, and amoxicillin (Table 3). They were named incorrectly in most instances.

### 2.2. Attitudes

#### 2.2.1. General

Pharmacists in this study unanimously agreed that there is currently abuse of antibiotics (104 (100%)). See Table 4.

#### 2.2.2. Experience and Sector

When the statement “Antibiotic resistance is a problem in Trinidad and Tobago” was compared with the sector of employment (*p* = 0.021). This was mainly in the private sector.

In reference to specific attitudes toward dual registration under the Food and Drug and Antibiotic Acts. The responses are displayed in Table 5. 

#### 2.2.3. Sector

With regards to the field-tested finding that “registration of antibiotics under the Food and Drug Act allows pharmacists to give patients antibiotics over the counter in the private setting”, there was a significant relationship between the private and public sectors (*p* = 0.013). This indicated that some pharmacists believed that registration of antibiotics under the Food and Drug Act allowed over-the-counter (OTC) dispensing. There was also a significant relationship seen between the responses to the statement “All antibiotics should be under the Antibiotic Act only” and the private sector (*p* = 0.002). A significant relationship existed between the attitude that registration under the Food and Drug Act did not allow over-the-counter dispensing and the attitude that all antibiotics should be registered under the Antibiotic Act only.

### 2.3. Practices

All responses in the area of practice showed no significant relationships concerning age. The results of these practices are displayed in Table 6.

#### Sector

The responses to the statement “Dispensing of antibiotics to patients over the counter in the private sector” showed a significant association with their sector of employment (*p* = 0.013). This association was important because it showed that pharmacists in this study significantly agreed that antibiotics were being dispensed OTC in the private sector. Most of the responding pharmacists were from the private sector. Their responses to the statements “A presenting patient is always asked to get a doctor’s advice before taking antibiotics”, “A presenting patient is always asked to get a doctor’s prescription before dispensing antibiotics”, and “Antibiotics are not dispensed over the counter without getting a doctor’s advice” also showed a significant association with their sector of employment, with *p*-values (*p* =0.007), (*p* = 0.008), and (*p* = 0.029) obtained, respectively. 

The pharmacists’ responses to the statements “Dispensing quinolones or sulfur drugs over the counter in the public sector” and “Dispensing quinolones or sulfur drugs over the counter as it is lawful” also showed significant relationships when compared to their sector of employment, with *p*-values (*p* = 0.016) and (*p* = 0.008) being obtained, respectively. Finally, when observing the responses to the statement “Dispensing of quinolones or sulfur drugs over the counter as it is profitable”, there was a significant association seen with the pharmacists’ sector of employment (*p* = 0.039). The results for the field practices showed a very significant relationship between the responses given and their sector of employment, particularly given that many of the responses supporting OTC dispensing were from those in the private sector.

### 2.4. Open-Ended Answers

Six main themes emerged from the questions that sought to group the answers to the second open-ended question in an ordered manner: 1. Dispensing with Prescription Only; 2. Allowance in Legislation for Legal OTC Dispensing; 3. Lack of Regulatory Enforcement; 4. Dispensing of Antibiotics OTC in Special Circumstances; 5. Doctors as Dispensers; 6. Suitcase Trade. 

Some of these important themes will be discussed below, which are displayed in Figure 1. Notably, the theme “Lack of regulatory enforcement” was the most common, followed by “Dispensing with prescription only under one Act”. 

Quinolones include ciprofloxacin, levofloxacin, and moxifloxacin. Aminoglycosides include gentamycin, while macrolides include clarithromycin and azithromycin.

Figure 2 shows the various antibiotic names, including quinolones, cefuroxime, and co-amoxiclav. One response stated: “There are no antibiotics under the Food and Drug Act”.

## 3. Discussion

When the various demographic factors were analyzed with the knowledge, attitudes, and practices of pharmacists in the study, it was found that:

### 3.1. Knowledge

The results revealed good knowledge of general issues surrounding antimicrobial resistance (AMR) and antibiotic resistance (AR) among pharmacists. Ventola (2015) previously noted that antibiotic resistance and the misuse of antibiotics are serious problems facing the world [5]. The author further notes that there are increased burdens on individuals due to lost salaries and the increased cost of health services [5]. This good general knowledge is thus a good start for antibiotic stewardship efforts. 

Knowledge was statistically tested against age, and 99% of respondents were significantly knowledgeable that the duration of hospitalization is affected if a resistant bug must be treated, with the majority being in the younger age group. Notably, at least one study from Trinidad and Tobago supports this knowledge that resistant bacteria lead to an increased duration of hospitalization [15]. The younger respondents were also significantly more knowledgeable about the fact that repeated use of the same antibiotics results in resistance. 

Gajdács et al. (2020), similar to our study, indicated that older pharmacists were “less confident” in their knowledge of inappropriate antibiotic use [16]. Kosiyaporn et al. (2020) indicated that “awareness of antibiotic use” and the resulting impact of AMR have been useful in “designing interventions” to combat AMR and inappropriate antimicrobial use (AMU) [17]. Thus, the fact that the pharmacists were knowledgeable about inappropriate use is a good starting point for designing communication and health promotion strategies. 

Voidăzan et al. (2019) showed that pharmacists were a major source of information on antibiotic resistance [18]. This information can be used strategically in Trinidad and Tobago to inform patients and combat inappropriate use [18]. Thus, the fact that younger pharmacists and, more importantly, pharmacists across all sectors are knowledgeable about these important issues is encouraging for future planning and intervention. 

#### Knowledge about the Antibiotic and Food and Drug Acts 

With regards to specific knowledge about the Antibiotic Act and the Food and Drug Act of Parliament, the responses showed variations in knowledge with experience and sex. Pharmacists with more experience had significantly more knowledge about antibiotics under the Food and Drug Act and Antibiotic Acts. With regards to knowledge of antibiotics registered under the Food and Drug Act, 84 (80.8%) respondents were aware that antibiotics are registered under this Act. Additionally, with regards to antibiotics registered under the Food and Drug Act and knowledge of the purview of the Antibiotic Inspectorate/Drug Inspectorate, 59 (56.7%) were knowledgeable. Thus, the majority of respondents were knowledgeable of the Food and Drug Act and the Antibiotic Act. Alternatively, 88 (84.6%) of respondents were knowledgeable about the Pharmacy Board Act and its role in antibiotic regulation.

Females were significantly more likely to be aware that some antibiotics are also registered under the Food and Drug Act, not just the Antibiotic Act. There is no specific comparison between sex and legislation in Trinidad and Tobago. Pham-Duc and Sriparamananthan (2021) and Zahreddine et al. (2018), however, have shown an association between greater knowledge and female sex [19,20].

Notably, the legal definition of antibiotics in the Food and Drug Act compared to the Antibiotics Act leads to the legal interpretation that there are no antibiotics registered in the Food and Drug Act [21]. This is supported by the open-ended statement from pharmacists that “there are no antibiotics under the Food and Drug Act”.

In completing the first open-ended question, some pharmacists incorrectly named antibiotics registered under the Food and Drug Act. Interestingly, some pharmacists named antibiotics under the Antibiotic Act as being under the Food and Drug Act, such as co-amoxiclav and cefuroxime. This highlights an important area for the education of pharmacists.

### 3.2. Attitudes

Pharmacists with more experience and working in all sectors significantly displayed the attitude that antibiotic resistance is a problem in Trinidad and Tobago, with 79.8% of respondents agreeing. The pharmacists were also knowledgeable about the fact that this is a worldwide problem, including the abuse of antibiotics. Thus, knowledge and attitudes about antibiotic resistance are well established in this study. 

The majority of pharmacists significantly disagreed that the registration of antibiotics under the Food and Drug Act allowed over-the-counter dispensing in the private sector (61.5%). However, 38.5% of pharmacists working in all sectors still agreed that registration of antibiotics under the Food and Drug Act does not lead to over-the-counter dispensing in the private sector. This question may have been sensitive, with possible legal repercussions perceived, and thus there may have been bias. 

The majority of pharmacists in both the private and public sectors significantly agreed (64.4%) that all antibiotics should be under the Antibiotic Act only. This highlights the possibility of the dichotomous laws contributing to the perceived inappropriate dispensing and may make antibiotic regulation more complex and costly by having two separate divisions responsible for different antibiotics [6]. This is a complex and ‘wicked’ issue of dichotomy and legal regulations.

Mahmoud et al. (2018) have highlighted the regulatory issues that exist in Saudi Arabia even after the introduction of new legislation [22]. Mate et al. (2019) have highlighted the issue of poor enforcement and weak inspection of pharmacies in Mozambique, and Adhikari et al. (2021) have highlighted regulatory issues and over-the-counter dispensing in Nepal [23,24]. Similar regulatory issues exist internationally; however, none of these countries highlights a dichotomy of laws as in Trinidad and Tobago. The study from Saudi Arabia also highlights that changing a law may not be a short- or medium-term solution [22]. 

#### Open-Ended Response supporting Attitudes

Alkadhimi et al. (2021), in their study in Iraq, found that pharmacists believed that they “could dispense antibiotics OTC for diarrhea and tonsillitis; they believed that they could be given the leeway to dispense for emergencies also” [25]. This is similar to the findings in the open-ended questions where pharmacists believe that, as professionals, they should be given a certain amount of autonomy to dispense antibiotics OTC under special circumstances and is congruent with the theme “pharmacists should be allowed to dispense antibiotics OTC in special circumstances”. 

Notably, the issue of enforcement of current legislation was highlighted in the statement that “the relevant Acts need to be revaluated and enforced because presently there is no enforcement of any rules and regulations”.

An issue in Trinidad and Tobago may be a lack of funding to adequately enforce prescriptions for appropriate antibiotic dispensing. The inspection role may also be inadequately staffed or funded [6,26]. 

### 3.3. Practices

#### 3.3.1. Age

Within the private sector, the practice of dispensing quinolones or sulfur drugs over the counter was performed because it was considered profitable by most respondents. Additionally, younger respondents were less likely to dispense antibiotics for profit. Notably, the majority of respondents were younger than 40, indicating that there may have been a bias towards younger respondents. They were also more likely to be in a public practice where OTC dispensing is prohibited or were less likely to be owners of the pharmacy. In studies in Lebanon and Saudi Arabia, they stated, as mentioned previously, that pharmacists with greater years of experience were inclined to dispense over the counter, and additionally, those pharmacists between 30 and 35 years of age were less likely to dispense over the counter [20,27]. 

Torres et al. (2020) highlighted the “driver of profits” and its influence on owners of pharmacies in Mozambique. This may be similar in the private sector in Trinidad and Tobago [28].

#### 3.3.2. Sector

Pharmacists dispense antibiotics over the counter in the private sector for profit and because of the perception that it is lawful, but not in the public sector. Notably, in the RHAs, the dispensing of antibiotics OTC is not allowed by policy. This allows accountability for the antibiotics in the public sector, inclusive of the five (5) Regional Health Authorities. Pharmacists at times failed to follow a doctor’s advice, and some dispensed without a prescription. Pharmacists also dispensed quinolones and sulfur-based antibiotics over the counter, which are registered under the Food and Drug Act. This corroborates the established narrative in this study, that over-the-counter dispensing does occur, and the finding that private sector pharmacists believe that dual registration under the two Acts may be contributing to the misuse of antibiotics. However, all pharmacists displayed the attitude that all antibiotics should be covered by one Act of parliament. 

Ventola (2015) further stated that the regulation of antibiotics is an issue in many countries, with over-the-counter (OTC) antibiotics being easily available. The OTC dispensing of antibiotics has been discussed in the introduction for Hungary, Tanzania, and Nepal [5]. Additionally, “antibiotics are thus accessible, plentiful, and cheap, which promotes overuse” in some countries, with antibiotics also being available online [5]. 

Alkadhimi et al. (2021) stated that pharmacists in Iraq had good knowledge of antimicrobial resistance but were still inclined to practice OTC dispensing, similar to this study [25]. In this study, it was uncovered that OTC dispensing does occur, and pharmacists had good knowledge and attitudes; however, their practices appear unchanged despite good knowledge. Thus, efforts must be made to change these practices, norms, and beliefs.

According to the Antimicrobial Resistance Collaborators (2022), 50,000 to 100,000 deaths in 2019 were attributed to bacteria resistant to antibiotics, such as quinolones [29]. Ayukekbong et al. (2017) emphasized that regulatory issues are a contributor to antimicrobial resistance [30]. In this study, these also appear to be similar practices. Interestingly, this study revealed that there is a similar problem with quinolones and sulfur-based antibiotics. 

Malik and Bhattacharyya (2019) stated previously that with their model, “lack of awareness can also accelerate the emergence of resistant strains and impart a significant economic cost on the population” [31]. This is driven by the irrational use of antibiotics in the community, as stated before [31]. Conversely, the evidence of a concerning lack of knowledge is highlighted in the statement that “dual registration allows for lawful dispensing of some antibiotics without a prescription, but it should still be controlled and not dispensed freely because of the monetary benefit”. This belongs to the theme of “belief that laws allow legal OTC dispensing” and again highlights the belief that it is legal to dispense certain antibiotics OTC. Thus, from the study, it can be deduced that the knowledge gap may be due to a misunderstanding of the regulatory issues in the Acts [13,14].

The practice in Trinidad and Tobago of dispensing quinolones and sulfur-based antibiotics because pharmacists perceive it to be lawful is indicative of the need for education. It is also congruent with findings from Jordan, Tanzania, Mozambique, Saudi Arabia, Iraq, Uganda, and Nepal that while regulations exist to control the dispensing of antibiotics, inappropriate dispensing patterns persist, such as in Trinidad and Tobago [22,23,24,25,32,33,34]. Similarly, Gebretekle and Serbessa’s (2016) paper found that fluoroquinolones and sulfur-based antibiotics, such as ciprofloxacin and co-trimoxazole, were the most frequently prescribed and possibly dispensed antibiotics [35]. Saleem et al. (2021) found high usage of co-amoxiclav and ciprofloxacin in the community. This is interesting, as pharmacists named co-amoxiclav as an incorrect drug under the Food and Drug Act. 

Gebretekle and Serbessa (2016) agreed that the practice of OTC dispensing is fueled by the need for profits, similar to the finding in our study [35]. Notably, there should be consideration for penalties or health promotion advice against errant behavior so as not to reinforce that behavior. Interestingly, similar to this study, Ayukekbong et al. (2017) indicated that regulatory factors drive antimicrobial resistance. They also indicated that community pharmacies offered unauthorized clinical consultations that suggested a diagnosis and may have contributed to unregulated dispensing [30]. The study also uncovered that pharmacists sometimes dispense without a prescription or doctors’ advice and sometimes do not ask the patient to get a doctor’s advice, uncovering similar issues. 

#### 3.3.3. Open-Ended Response Supporting Practices

The concept of illegal trade and antibiotic misuse was highlighted in the statement that the “availability of antibiotics via suitcase traders is a problem” under the theme of ‘Suitcase Trade’. The issue of the suitcase trade of pharmaceuticals in Trinidad and Tobago has been highlighted previously [36]. This practice would be a breach of existing regulations. It is not directly related to OTC dispensing but may contribute by providing substandard medication. 

It has been recommended that suppliers from different countries be approved, not just locally based suppliers. Foreign-based suppliers should be considered for registration. Additionally, companies should not have to use only intermediate distributors, as is the current practice. This helps expand access to cheaper drugs [37]. 

Kakkar (2020) highlights that factors hampering adequate or effective regulation include a lack of political will and relaxed regulatory mechanisms [26]. Thus, the errant practices could be fueled by regulatory deficiencies [26]. Kurdi et al. (2020) have shown that even with new antibiotic regulations for controlling the dispensing of antibiotics in Saudi Arabia, the practice of OTC dispensing persists [34]. The authors further state that “educational programs and campaigns” are recommended. Jacobs et al. (2019) also recommended a multi-faceted approach to tackle the inappropriate dispensing of antibiotics in a review of OTC sales of antibiotics. Thus, changing the dichotomy of laws may not be the only answer [38]. 

The practice of pharmacy assistants and other categories of staff other than pharmacists dispensing prescription drugs to patients also breaches national laws [32]. This is highlighted under the theme “suitcase trade” in the statement, “A lot of pharmacies dispense antibiotics without a pharmacist in the pharmacy and also operate without a pharmacist present in the pharmacy.” This supports the point that OTC dispensing practices may be facilitated by giving staff other than pharmacists the authority to dispense [32].

### 3.4. Limitations

There may have been some degree of apprehension in the responses to the questionnaires that were administered with assistants compared to those that were self-administered. This also led to removing the requirement for emails. This removal meant that the “only one response per participant” feature in Google Forms was removed. This may have led to duplicate responses and bias. The non-structured method of collecting data, including the convenience method of collecting data, means that the findings cannot be generalizable to the population. It was uncovered, however, that a problem with OTC dispensing does exist in Trinidad and Tobago. Some of the causes have been identified in this study.

### 3.5. Conclusions

Pharmacists appear to have significant and encouraging knowledge and attitudes regarding antimicrobial resistance and the effects of inappropriate dispensing of antibiotics. Over-the-counter dispensing still occurs in the private sector in Trinidad and Tobago. Additionally, practices were not congruent with positive knowledge and attitudes. 

This is the first study examining the knowledge, attitudes, and practices of pharmacists, especially concerning over-the-counter dispensing with the two Acts of Parliament. This study is also unique in that it examines dispensing in a country with two laws governing antibiotic dispensing, inspections, and thus regulation. It uncovered that there was a deficiency in understanding the role of the various regulatory issues under each Act and that this dichotomy of laws may be contributing to inappropriate practices.

## 4. Materials and Methods

A quantitative, cross-sectional study was conducted. The study targeted pharmacies across the public and private sectors in Trinidad. Two populations were targeted in the study: public-sector pharmacists and pharmacists in the private sector. 

In Trinidad and Tobago, public sector health services are split into five self-managed regions known as Regional Health Authorities (RHA). They are governed by the Regional Health Authorities Act, Chapter 29:05, Act 5 of 1994. (Ministry of The Attorney General and Legal Affairs, 2016). 

The study was conducted from April to October 2021, over a seven-month period. 

### 4.1. Study Sample

A stratified sampling of pharmacists from the Sangre Grande Hospital (SGH), St Andrews/St David (STAD), Nariva/Mayaro (NAMA), and private sector pharmacies was performed to obtain a sample of pharmacists from the public and private sectors. 

The study initially targeted 145 pharmacists (across public and private-sector pharmacies); however, 104 were recruited from both the private and public sectors (~72% response rate). The public sector consisted of approximately 250 pharmacists out of a total of 641 pharmacists nationally (6). One RHA (the Eastern RHA) with 45 pharmacists was purposely selected to represent the public sector responses for this study. All the public-sector pharmacists at the Eastern RHA participated in the study. They represented pharmacists for: Biche Outreach Centre, Black Rock Outreach Centre, Brothers Road Outreach Centre, Coryal Outreach Centre, Cumuto Outreach Centre, Grande Riviere Outreach Centre, Guayaguayare Outreach Centre, Manzanilla Outreach Centre, Matelot Outreach Centre, Matura Outreach Centre, Mayaro District Health Facility, Rio Claro Health Centre, San Souci Outreach Centre, Sangre Grande Enhanced Health Centre, Toco Health Centre, 24-h Accident and Emergency, the Valencia Outreach Centre, and the Sangre Grande Hospital. All public sector pharmacies and thus pharmacists follow the same regulations for public sector workers, which is why we chose a sample from one RHA rather than all. The COVID-19 pandemic also hindered movement and was a major consideration for the study design. There are approximately 375 registered private pharmacies in the country. Initially, the study sought to recruit 100 pharmacists from the private sector. It was noted that pharmacists in the private sector may work in public institutions and vice versa. A 95% confidence level was used, and a margin of error of 8.4% was assumed for this calculation. The response distribution was assumed to be 50%. This was performed because responses were obtained during a period of COVID-19 restrictions and due to the legal sensitivity of the topic.

### 4.2. Recruitment and Inclusion Criteria 

Public Sector—All pharmacists with at least one year of dispensing experience in the in-patient and out-patient departments of the Sangre Grande Hospital, St. Andrews/St. David County Health Administration, and Nariva/Mayaro County Health Administration were eligible for inclusion in the survey and were given an equal chance of participation. Participation was voluntary, and all eligible public sector pharmacists in the inpatient and outpatient departments were offered a questionnaire from April 2021 to October 2021. Participants had the option of choosing the public sector or both the private and public sectors (they used the option “both” to do this). They chose both if they were captured in a public setting but worked privately also, and additionally if they were captured in a private setting but worked publicly also. No duplication was allowed when visiting pharmacies.

The survey was administered via email to the senior pharmacists at each facility. These senior pharmacists then distributed the survey via email to their junior colleagues. The respondents accessed the consent form and survey via a link using Google Forms (settings were adjusted to “responses required”). 

Private Sector: The private pharmacies were engaged via referral from colleagues and encompassed pharmacies throughout the country. Pharmacists were recruited by snowballing in the private sector. Three pre-trained assistants helped with administering the questionnaire via telephone, gathering data, and recruiting participants by snowballing (for the private sector pharmacists). Participation in the study was also voluntary.

### 4.3. Data Collection

The questionnaire was administered via Google Forms, and data was collected using a structured questionnaire with two open-ended questions. Data were collected for 7 months (up until the point of saturation and no more referrals from colleagues) from pharmacists in the private and public sectors. 

Participants were emailed the Google Form (settings were adjusted to “responses required”). In some cases, WhatsApp private messenger was used, and a link to the Google form was sent to the respondents. Initially, participants were sent Google Forms with a feature for the collection of email addresses. This was removed a month into the study to improve the confidentiality of the respondents and increase confidence in the anonymity of the study. 

### 4.4. Exclusion Criteria

Persons employed at pharmacies who were not pharmacists and pharmacists with less than one year of dispensing experience were excluded from the study.

### 4.5. Data Analysis

The collected data was inputted and analyzed using IBM SPPSv22 and Microsoft Excel software. Descriptive statistics were used in analyzing the data collected. The Chi-squared (X^2^) test was used to measure the observed and expected variables of the categorical demographic variables versus the questioner’s items. The study looked at whether relationships existed between age, sex, years of experience, or sector and the participant’s knowledge, attitudes, and perceptions based on the questions asked in the questionnaire. This assumes that there are dependent and independent variables, and they are categorical. The significance level was set at *p* < 0.05 for all tests. Fisher’s exact *t*-test was also used for fields with numbers less than or equal to 5. We avoided adjusting further for alpha to avoid Type 2 errors (39). Ranganathan (2016) further confirms that studies should not rely overly on alpha adjustments, as this may also lead to erroneous results. This was also given that we accepted an increased margin of error due to the sensitive nature of the study and the COVID-19 environment. Instead, we focused on the primary outcome of the presence or absence of over-the-counter dispensing (39). Two open-ended questions were used. These were: “Can you name an antibiotic under the Food and Drug Act?” and “Is there anything else you would like to say about dual registration of antibiotics and dispensing? Open-ended responses to the second question were analyzed and grouped into emerging themes. 

### 4.6. Privacy and Confidentiality of Participant Information and Research Data

All participant identifiers were removed. No participant’s personal information was required for this study, nor was patient information needed. To ensure privacy, no names were collected. Only electronic forms with no identifiers were used. Informed consent was obtained. The consent forms contained no addresses, email addresses, or phone numbers that could be used to trace the responses back to specific participants. Any personal data obtained from the use of WhatsApp and Google Forms was stored on a password-protected computer, with any hardcopy information kept in a locked cupboard. Most of the data for this study was stored electronically on a password-protected computer.

### 4.7. Ethical Considerations

Ethical permission was sought and granted from the University of Trinidad and Tobago (UTT) and the Eastern Regional Health Authority’s (ERHA) Ethics Committee.

## Figures and Tables

**Figure 1 antibiotics-12-01094-f001:**
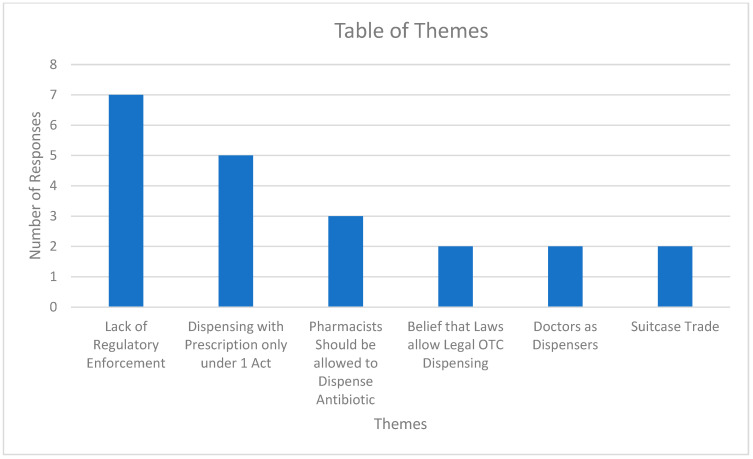
Graph of Themes for Open-Ended Questions. “Suitcase trade” refers to the illegal trade of pharmaceuticals.

**Figure 2 antibiotics-12-01094-f002:**
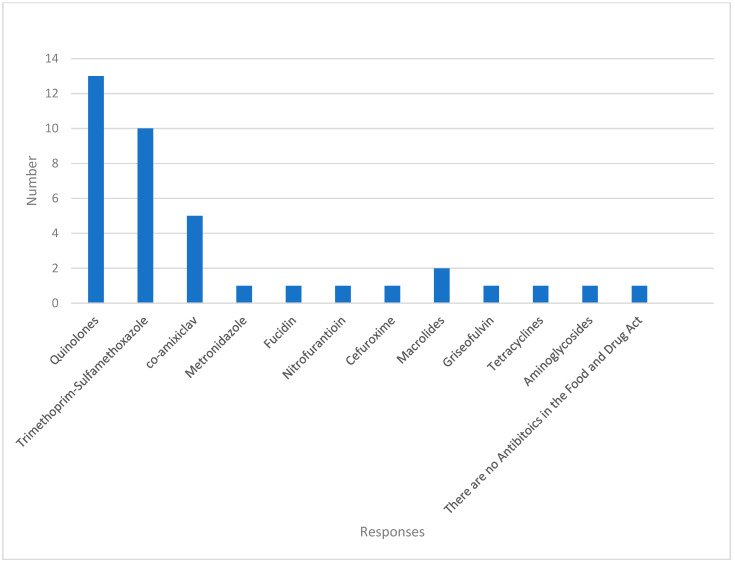
Responses to the Naming of Antibiotics under the Food and Drug Act.

**Table 1 antibiotics-12-01094-t001:** Significance tests showing the relationship between Knowledge of Antibiotic Resistance and the age, sex, experience, and sector of participants.

Knowledge	Yes n (%)	No n (%)	*p*-Values ^+^			
			Age	Sex	Experience	Sector
Antibiotic Resistance is associated with inappropriate antibiotic use.	103 (99)	1 (1)	0.578	0.404	0.221	1.00
Repeated use of the same antibiotics results in resistance.	97 (93.3)	7 (6.7)	0.027 *	0.349	0.057	0.158
Antibiotics can speed up recovery from the flu or the common cold.	15 (14.4)	89 (85.65)	0.271	0.974	0.616	0.668
Antimicrobial Resistance results in Resistance to Antibiotics only.	32 (30.8)	72 (69.2)	0.360	0.206	0.255	0.017 **
Antibiotic resistance is a serious public health problem facing the world. ***	104 (100)	0 (0)	n/a	n/a	n/a	n/a
Antibiotics can speed up the recovery of people suffering from COVID-19.	23 (22.1)	81 (77.9)	0.446	0.349	0.870	0.684
Superbugs, such as MRSA and carbapenem-resistant Gram-negative bacilli, result in fewer antibiotic choices.	97 (93.3)	7 (6.7)	0.587	0.349	0.684	0.649
Superbugs, such as MRSA and carbapenem-resistant Gram-negative bacilli, result in increased costs.	94 (90.4)	10 (9.6)	0.125	0.515	0.173	0.760
Resistant Gram-negative bacilli result in an increased length of stay for patients on wards.	103 (99)	1 (1)	0.001 *	0.596	1.00	1.00
Vaccines can prevent unnecessary antibiotic use and, thus, antibiotic resistance.	70 (67.3)	34 (32.7)	0.130	0.112	0.440	0.001 **

^+^ *p* < 0.05 is considered significant. The Chi^2^ test was used to test the significance of relationships. Fisher’s Exact *t*-test was used where cells have small values (less than or equal to 5). * Younger pharmacists were more knowledgeable than older pharmacists with regards to repeated use of antibiotics and increased length of stay on the wards. ** Pharmacists in both the private and public sectors significantly believed that antimicrobial resistance is not only to antibiotics, but more private sector pharmacists knew that vaccines could prevent unnecessary antibiotic use. *** All pharmacists agreed that antibiotic resistance is a serious worldwide problem.

**Table 2 antibiotics-12-01094-t002:** Knowledge of the Antibiotic Act and Pharmacy Board Act.

Knowledge about the Antibiotic and Pharmacy Board Acts	Yes n (%)	No n (%)	*p*-Values ^+^			
			Age	Sex	Experience	Sector
The Pharmacy Board Act regulates the dispensing of antibiotics by pharmacists.	88 (84.6)	16 (15.4)	0.644	0.798	0.817	0.321
Have you heard of the Antibiotic Act?	101 (97.1)	3 (2.9)	0.146	1.00	0.827	1.00
Have you heard of the Antibiotic Inspectorate/Drug Inspectorate?	97 (93.3)	7 (6.7)	0.360	0.510	0.726	0.262

^+^ *p* < 0.05 is considered significant. The Chi^2^ test was used to test the significance of relationships. Fisher’s exact *t*-test was used where cells had small values (less than or equal to 5).

**Table 3 antibiotics-12-01094-t003:** Knowledge of the Food and Drugs Act and Antibiotic Act.

Knowledge About Food and Drug Act	Yes n (%)	No n (%)	*p*-Values ^+^			
			Age	Sex	Experience	Sector
Are All antibiotics registered under the Antibiotic Act?	31 (29.8)	73 (70.2)	0.461	0.507	0.374	0.398
Are any Antibiotics registered under the Food and Drug Act?	84 (80.8)	20 (19.2)	0.489	0.047 *	0.128	0.084
Are Antibiotics resisted under the Food and Drug Act under the purview of the Antibiotic Inspectorate/Drug Inspectorate?	59 (56.7)	45 (43.3)	0.051	0.254	0.036 **	0.509
Can you name an antibiotic registered under the Food and Drug Act?	73 (70.2)	31 (29.8)	0.292	0.128	0.222	0.141

^+^ *p* < 0.05 is considered significant. This is denoted in red and underlined. The Chi^2^ test was used to test the significance of relationships. Fisher’s exact *t*-test was used where cells had small values (less than or equal to 5). * Female pharmacists significantly knew that antibiotics are also registered under the Food and Drugs Act. ** More experienced pharmacists significantly knew that antibiotics are resisted under the Food and Drug Act under the purview of the Antibiotic Inspectorate/Drug Inspectorate.

**Table 4 antibiotics-12-01094-t004:** Attitudes to Antibiotic Dispensing.

Attitudes	Agree n (%)	Disagree n (%)	Do Not Know	*p*-Values ^+^			
				Age	Sex	Experience	Sector
Antibiotics should be given to patients when they ask for them without a prescription.	3 (2.9)	101 (97.1)	0 (0)	0.216	0.356	1.00	0.320
When I have a cold or flu, antibiotics help me get better.	12 (11.5)	92 (88.5)	0 (0)	0.783	0.248	0.355	0.702
Antibiotics should be stopped as soon as a person feels better, not after the recommended course.	2 (1.9)	102 (98.1)	0 (0)	0.392	0.353	1.00	0.277
Skipping antibiotic doses does not contribute to resistance.	9 (8.7)	95 (91.3)	0 (0)	0.882	0.795	0.581	0.575
Antibiotic resistance is a problem in Trinidad and Tobago.	83 (79.8)	1(1)	20 (19.2)	0.086	0.881	0.318	0.021 *
There is currently an abuse of antibiotics. **	104 (100)	0 (0)	0 (0)	n/a	n/a	n/a	n/a
The COVID-19 pandemic has worsened the problem of antibiotic abuse.	87 (83.7)	17 (16.3)	0 (0)	0.570	0.249	0.468	0.776
The public and I should vaccinate to avoid unnecessary antibiotic use.	71 (68.3)	33 (31.7)	0 (0)	0.518	0.63	0.692	0.102

^+^ *p* < 0.05 is considered significant. This is denoted in red and underlined. n/a means not applicable, as the response was 100%, and thus all pharmacists answered affirmatively. The Chi^2^ test was used to test the significance of relationships. Fisher’s exact *t*-test was used where cells had small values (less than or equal to 5). * Pharmacists, mainly in the private sector, believed that antibiotic resistance was a problem. ** Pharmacists, mainly in the private sector, agreed that there is abuse of antibiotics.

**Table 5 antibiotics-12-01094-t005:** Percentages and Chi Square tests showing the relationships between Attitudes towards Dual Registration and age, sex, experience, and sector.

Attitudes to Dual Registration	Agree n (%)	Disagree n (%)	*p*-Values ^+^			
			Age	Sex	Experience	Sector
Registration of Antibiotics under the Food and Drug Act and the Antibiotic Act is good.	76 (73.1)	28 (26.9)	0.681	0.755	0.804	0.416
Registration of Antibiotics Under the Food and Drug Act, pharmacists are allowed to give patients Antibiotics over the counter in a Public setting.	10 (9.6)	94 (90.4)	0.910	0.167	0.140	0.300
Registration of Antibiotics Under the Food and Drug Act, pharmacists are allowed to give patients Antibiotics over the counter in a private setting.	40 (38.5)	64 (61.5)	0.290	0.636	0.285	0.013 *
All antibiotics should be under the Antibiotic Act only.	67 (64.4)	37 (35.6)	0.090	0.981	0.241	0.002 *

^+^ *p* < 0.05 is considered significant. This is denoted in red and underlined. The Chi^2^ test was used to test the significance of relationships. Fisher’s exact *t*-test was used where cells had small values (less than or equal to 5). * Pharmacists (38.5%), mainly in the private sector, significantly believed that having antibiotics under the Food and Drug Act allowed pharmacists to give patients antibiotics over the counter in the private setting and that all antibiotics should be under one Act.

**Table 6 antibiotics-12-01094-t006:** Practices of Pharmacists towards Antibiotic Dispensing and Chi Square tests comparing the relationships with age, sex, experience, and sector of work.

Practices	Never n (%)	Sometimes n (%)	Always n (%)	*p*-Values ^+^			
				Age	Sex	Experience	Sector
Dispensing of antibiotics to patients over the counter in the private sector.	71 (68.3)	32 (30.8)	1 (1)	0.532	0.313	0.410	0.013 *
A presenting patient is always asked to get a doctor’s advice before taking antibiotics.	1 (1)	26 (25)	77 (74)	0.344	0.454	0.693	0.007 *
A presenting patient is always asked to get a doctor’s prescription before dispensing antibiotics.	1 (1)	28 (26.9)	75 (72.1)	0.105	0.543	0.858	0.008 *
Antibiotics are not dispensed over the counter without getting a doctor’s advice.	13 (12.7)	31 (30.4)	58 (56.9)	0.504	0.305	0.055	0.029 *
Dispensing quinolones or sulfur drugs over the counter in the public sector.	94 (90.4)	9 (8.7)	1 (1)	0.620	0.690	0.674	1.00
Dispensing quinolones or sulfur drugs over the counter in the private sector. *	40 (38.5)	55 (52.9)	6 (5.8)	0.065	0.590	0.988	0.016 *
Dispensing quinolones or sulfur drugs over the counter as it is lawful.	45 (43.3)	52 (50)	7 (6.7)	0.475	0.783	0.981	0.008 *
Dispensing quinolones or sulfur drugs over the counter as it is profitable.	73 (70.2)	28 (26.9)	3 (2.9)	0.062	0.623	0.340	0.039 *

^+^ *p* < 0.05 is considered significant. This is denoted in red and underlined. The Chi^2^ test was used to test the significance of relationships. Fisher’s exact *t*-test was used where cells had small values (less than or equal to 5). * The private sector pharmacists were significantly associated with responses to over-the-counter dispensing and other possibly errant practices, such as dispensing without a prescription or doctors’ advice.

## Data Availability

Data sharing not applicable. No new data were created or analyzed in this study. Data sharing is not applicable to this article.

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
