# Peer review of "Knowledge, Attitudes, and Practices in Antibiotic Dispensing amongst Pharmacists in Trinidad and Tobago: Exploring a Novel Dichotomy of Antibiotic Laws"

_antibiotics, 2023, doi:10.3390/antibiotics12071094_

Round 1
Reviewer 1 Report
Dear authors, I reviewed your paper and have the following comments
MAJOR
- it is not clear how your sample is representative for the general population of pharmacists in your country. Please elaborate
-the sample size is relatively low and the issue of statistical significance is questionable in my view. I am not a statistician but this looks like multiple testing which needs to be predefined and controlled, otherwise the alpha is multiplied and you get some of the positive results by chance
-some of the questions related to knowledge about AMR are ambiguous in my view. Examples: repeated use of the same antibiotic leads to resistance or antibiotics can speed up recovery from COVID-19. (every use of antibiotics can generate resistance, repeated treatment is not the same if repeated after years, COVID-19 can have a superinfection and then an antibiotic speeds up recovery)
In my view, the above only do not allow your paper to be published. I would have had other comments as well, but the above seem like serious flaws.
-lines 25-26: the percentages given do not add up to 100%.
Minor
-line 17-18: a pattern cannot be higher, please reword
-lines 18-19: accesibility to what? please reword
Author Response
MAJOR
it is not clear how your sample is representative for the general population of pharmacists in your country. Please elaborate:
Response
- It was not meant to be representative and this is stated in the limitation
-
This study was conducted during the restrictions of the COVID-19 pandemic. It also dealt with a sensitive issue of possible infringements of a law. This statement is added at the end of the introduction to give context to the limitations.
- The method was designed to work within the restrictions of a pandemic and the low response expected if a person has to answer truthfully that a law is being broken. The low response was factored in by using a higher margin of error acceptance.
- This study also presented rich qualitative data in the form of open-ended thematic responses. In fact, qualitative studies are frequently published and have an inherent bias and are not generalizable. It does not mean they have no merit. The methodology was constructed to deal with the environment and sensitivity of the topic. it is also reproducible. We sought to find an endpoint of the presence or absence of aver the counter dispensing.
- The Fisher's exact t-test is also done to add internal statistical validity
-the sample size is relatively low and the issue of statistical significance is questionable in my view. I am not a statistician but this looks like multiple testing which needs to be predefined and controlled, otherwise the alpha is multiplied and you get some of the positive results by chance
- Change/ How addressed: The methodology reflects this now. The public sector consisted of approximately 250 pharmacists out of a total of 641 pharmacists nationally (6). One RHA (the Eastern RHA) with 45 pharmacists was purposely selected to represent the public sector responses for this study. All the public sector pharmacists at the Eastern RHA participated in the study. They represented pharmacists for: Biche Outreach Centre, Black Rock Outreach Centre, Brothers Road Outreach Centre, Coryal Outreach Centre, Cumuto Outreach Centre, Grande Riviere Outreach Centre, Guayaguayare Outreach Centre, Manzanilla Outreach Centre, Matelot Outreach Centre, Matura Outreach Centre, Mayaro District Health Facility, Rio Claro Health Centre, San Souci Outreach Centre, Sangre Grande Enhanced Health Centre, Toco Health Centre and 24 h Accident and Emergency, the Valencia Outreach Centre and the Sangre Grande Hospital. All public sector pharmacies and thus pharmacists follow the same regulations for public sector workers and hence the reason we chose a sample from one RHA rather than all. The COVID-19 pandemic also hindered movement and was a major consideration for the study design. There are approximately 375 registered private pharmacies in the country. Initially, the study sought to recruit 100 pharmacists from the private sector. It was noted that pharmacists in the private sector may work in public institutions and vice versa. A 95% confidence was used and a margin of error of 8.4% was assumed for this calculation. The response distribution was assumed to be 50%. This was done because responses were obtained during a period of COVID-19 restrictions and due to the legal sensitivity of the topic. The 8.4% margin of error was used for a smaller sample size.
- Fisher’s exact t-test was also used for fields with numbers less than or equal to 5 (to account statistically for small numbers).
- Addressing Multiple Testing in Statistics: Your reference is to an alpha adjustment. We avoided adjusting further for alpha to avoid Type 2 errors. Ranganathan (2016) further confirms that studies should not rely overly on alpha adjustments as this may also lead to erroneous results.
We hope this response can satisfy you as this is from valid statistical sources.
-some of the questions related to knowledge about AMR are ambiguous in my view. Examples: repeated use of the same antibiotic leads to resistance or antibiotics can speed up recovery from COVID-19. (every use of antibiotics can generate resistance, repeated treatment is not the same if repeated after years, COVID-19 can have a superinfection and then an antibiotic speeds up recovery)
- How Questions are Addressed: For repeated use it was explained to the pharmacists, if they asked, that this meant the same antibiotics within a short duration of the same course. With regards to COVID-19, this is a virus and while you can have bacterial superinfection you do not use this as your first line of treatment. This would have been explained to pharmacists if they asked.
The majority of these questions were derived from reputable journal sources and we believe that the majority of these questions are valid. You can see how comprehensive our references are.
These questions were derived from published papers and are accepted questions.
In my view, the above only doe not allow your paper to be published. I would have had other comments as well, but the above seem like serious flaws.
- Response: We think that our study has merit and deals with a phenomenon of a dichotomy of laws dealt with nowhere else in the world. The study was not meant to be representative of the population and the smaller sample size was factored in when calculating the margin of error. This study involved possible infractions of laws and was thus sensitive. It also took place during the limitations of movement of the COVID-19 pandemic. In fact, this study shows how a study can be done when looking at these sensitive topics under non-ideal situations. We looked at a primary endpoint of the presence or absence of over-the-counter dispensing as our main outcome. We just wanted to show that a problem existed. We even did the Fisher's test to add validity. Regression cannot be done as we are not dealing with numerical statistics by a Likert-type scale.
-lines 25-26: the percentages given do not add up to 100%.
- Change: The percentages were removed as these reflected those who were in ONLY private work or ONLY public work. Just over 23 % were in BOTH private and public practice. It was not meant to be 100%.
Minor
-line 17-18: a pattern cannot be higher, please reword
- Change: This pattern of inappropriate consumption differs in higher-income countries due to the ease of accessibility. please reword
-lines 18-19: accesibility to what?
- Change: "of antibiotics" has been added
Reviewer 2 Report
The manuscript aimed to assess the knowledge, attitudes, and practices of pharmacists regarding the over-the-counter dispensing of antibiotics in Trinidad and Tobago (T & T). The study gathered data from 104 pharmacists in both private and public sectors over a period of seven months.
The study revealed that antibiotic resistance and abuse were perceived as significant problems by (T & T) pharmacists. It appears that pharmacists with experience, female gender, and younger age showed better knowledge regarding antibiotics. (T & T) pharmacists believed that antibiotics regulated under the Food and Drug Act may have contributed to over-the-counter dispensing in the private sector, and all antibiotics should be regulated under the Antibiotic Act. In terms of practices, the study found that the dispensing of antibiotics over-the-counter in the private sector occurred without doctors' advice, without requesting prescriptions, and often due to the perception that it was lawful. Older pharmacists and the motivation for profit were factors influencing this practice.
The results exemplified deficient enforcement in pharmacy regulations with regard to prescribing and dispensing of antibiotics.
This discussion is very important given the urgent call to fight the problem of antimicrobial resistance globally.
However to improve the readability of the manuscript the authors need to follow the following advice.
-Age , experience, sector are demographic factors are repeated multiple times but these results could be consolidate and contextualized in one place.
-Instead of using only Chi-square test the authors should also consider logistic regression. Even for Chi-square test, there are tables with frequencies less than 5 where an appropriate statistical test would have been a Fischer test.
-Please add the Ethics statement indicating the ethical approval of the proposal by the Ethics Committee and whether pharmacists consented to participate in the survey.
-Why did you send Google Forms with a question collecting emails addresses of participants in the first place? this is violation of confidentiality!
- I suggest for figure 1 and 2 to arrange the bars in increasing frequency of responses for easy readability.
none
Author Response
Age , experience, sector are demographic factors are repeated multiple times but these results could be consolidate and contextualized in one place.
- Response: These demographic factors belong to the 3 sections separately. Combining them will confuse the data.
-Instead of using only Chi-square test the authors should also consider logistic regression. Even for Chi-square test, there are tables with frequencies less than 5 where an appropriate statistical test would have been a Fischer test.
- Response: The Fisher's Exact t-test is done. Regression analysis cannot be done on nominal variables in a Likert type scale.
-Please add the Ethics statement indicating the ethical approval of the proposal by the Ethics Committee and whether pharmacists consented to participate in the survey.
- Response: This is done
-Why did you send Google Forms with a question collecting emails addresses of participants in the first place? this is violation of confidentiality!
- Response; Thanks for the comment. Google Forms automatically collects emails but you can disable this feature. We disabled it. It is only a violation if you do not password-protect the data or store it in a secure location. We stored the data securely and on one computer. Thus we did not violate confidentiality or privacy (noting that both are different but were treated in a professional manner).
I suggest for figure 1 and 2 to arrange the bars in increasing frequency of responses for easy readability.
- Response: This was done as suggested. Thank you for this.

Reviewer 3 Report
Thank you for the opportunity to review this manuscript describing s well-designed and important survey study of pharmacist knowledge in Trinidad and Tobago regarding inappropriate antimicrobial dispensing and use. I have 2 concerns:
1. In Tables 1-6 of the Results section, please indicate in detail how the significant results differed. Each Table should be completely self explanatory. For example in Table 1, Age and Sector correlated significantly with 4 questions. I would suggest using superscripts next to the P values and including the definitions in the Table footnotes (eg., *younger pharmacists were more knowledgeable than older pharmacists; **public sector pharmacists were more knowledgeable than private sector pharmacists). This should be done in all 6 Tables, and in the paragraphs following each Table. You give this detail in the Discussion, but it should be included in the Results as well.
2. In Table 1, Experience was not a significant correlate for question 1 if the P value was equal to 0.05, and it should not be highlighted and underlined. In the Age and Experience paragraph following Table 3, you do state correctly that "Notable, for this field with regards to age, there was no significant association for age (p=0.05)."
3. The Limitations section should address your small sample size, and whether it is truly representative of the knowledge and attitudes of pharmacists throughout the country.
a. In the Materials and Methods section, you state Public Sector pharmacists included those with at least 1 year of experience at only 3 institutions in the Eastern RHA, with a total of 45 pharmacists. How many pharmacists in total are employed in the Public Sector in Trinidad and Tobago? Is your sample representative? Are pharmacist knowledge and attitudes consistent throughout all RHAs in the country (for example, is the Eastern RHA more urban or rural than other RHAs, etc.)?
a. For the Private Sector, there are approximately 375 registered pharmacies in the country. You targeted a total of 100 pharmacists, and received a total of 59 responses. How many pharmacists in total are employed in the Private Sector? Is your sample representative?
English is fine.
Author Response
In Tables 1-6 of the Results section, please indicate in detail how the significant results differed. Each Table should be completely self-explanatory. For example in Table 1, Age and Sector correlated significantly with 4 questions. I would suggest using superscripts next to the P values and including the definitions in the Table footnotes (eg., *younger pharmacists were more knowledgeable than older pharmacists; **public sector pharmacists were more knowledgeable than private sector pharmacists). This should be done in all 6 Tables, and in the paragraphs following each Table. You give this detail in the Discussion, but it should be included in the Results as well.
- Response: Thank you for this comment. This makes the table busy but was done as requested.
2. In Table 1, Experience was not a significant correlate for question 1 if the P value was equal to 0.05, and it should not be highlighted and underlined. In the Age and Experience paragraph following Table 3, you do state correctly that "Notable, for this field with regards to age, there was no significant association for age (p=0.05)."
- Response: Changed
3. The Limitations section should address your small sample size, and whether it is truly representative of the knowledge and attitudes of pharmacists throughout the country.
- Response: No it is not representative and we make provisions in accepting a larger margin of error in the statistical calculation. We tried to ascertain a common endpoint such as the presence or absence of over-the-counter dispensing. The sampling and its methods were not meant to be representative.
- Qualitative studies are published all the time and they are never representative. We also have "rich" open-ended questions. The study also deals with a novel topic of a dichotomy of laws contributing to the problem of antibiotic resistance. This is important data that was surveyed an a manner appropriate to the sensitivity of the topic and COVID-9 environment.
a. In the Materials and Methods section, you state Public Sector pharmacists included those with at least 1 year of experience at only 3 institutions in the Eastern RHA, with a total of 45 pharmacists. How many pharmacists in total are employed in the Public Sector in Trinidad and Tobago? Is your sample representative? Are pharmacist knowledge and attitudes consistent throughout all RHAs in the country (for example, is the Eastern RHA more urban or rural than other RHAs, etc.)?
- Response: This is elaborated on in the methodology. There are approximately 250 out of over 600 in total. The sample while not representative, due to the lack of randomization, gives an idea of the knowledge, attitudes, and practices as it pertains to the public sector. It tells us if there is a problem there. During this time of data collection, services were curtailed, there was limited movement and persons were being asked about a sensitive issue of a breach of the law. Using one RHA helped and was sensible and strategic, as all RHAs follow the same policies and practices with regard to dispensing. This is different from the private sector.
- The Fisher's exact t-test is also done to add internal statistical validity
a. For the Private Sector, there are approximately 375 registered pharmacies in the country. You targeted a total of 100 pharmacists, and received a total of 59 responses. How many pharmacists in total are employed in the Private Sector? Is your sample representative?
Response: We accepted an 8.5% Margin of Error to account for issues related to movement during the pandemic and also the low response expected as persons are responding to questions asking about beaches in-laws. It was a sensitive topic. This can be reproduced. The COVID-19 environment also made data collection difficult. Additionally, we were dealing with a legally sensitive issue of global importance.
Round 2
Reviewer 1 Report
Thanks for taking previous comments into account and for the improvements. I take the point that there is some merit in the paper, although I still believe its methodological quality is not what it could have been. I leave the editors to decide whether they would like to publish
Author Response
Thanks for taking previous comments into account and for the improvements.
- Thank you for considering our well-thought-out response to the comments.
I take the point that there is some merit in the paper, although I still believe its methodological quality is not what it could have been.
- Thanks for understanding the merits
- - With regards to the methodology, we have made the improvements by including:
- context of an expected low response
- The adjustment of the margin of error
- The fact that this was carried out during periods of limited mobility of COVID-19
- The use of Fisher's Exact t-test
I leave the editors to decide whether they would like to publish
- We believe that our paper should be published given the improvements
- That the reviewer agrees that there is merit
- This deals with a novel issue of the dichotomy of antibiotic laws. This has not been seen before in any literature. It can guide policymakers in understanding that dichotomy has not worked in one area of the world.
- We cannot sacrifice ethics and this methodology has been improved as much as ethically possible.
Reviewer 3 Report
Thank you for addressing my comments, but please allow me to clarify:
1. Tables 1-6 show the overall survey results (Yes or No, etc), and also the results of your Chi square (or Fisher exact test) comparing age, gender, experience, and sector (public or private). Any significant P value should be explained. It is not necessary to mark each P value to indicate Chi square or Fisher exact, because you described in Section 4.4 (Data analysis) that the "Fisher’s exact t-test was also used for fields with numbers less than or equal to 5."
a. In Table 1, Age was significant for questions 2 and 9. You have an (*) next to each P value, and below the Table you can just state "*Younger pharmacists were more knowledgeable than older pharmacists;".
b. In Table 1, Sector was significant for questions 4 and 10. You have a (**) next to each P value, but your caption below the Table states "**Pharmacists in all sectors significantly believed that antimicrobial resistance is not only to antibiotics and that vaccines can prevent unnecessary antibiotic use." This should be a comparison between public sector and private sector pharmacists. Which sector was more knowledgeable about question 4, and which was more knowledgeable about question 10?
c. In Table 4, Sector was significant for question 5. You have an (*) next to the P value, but again state below the Table "*Pharmacists in all sectors and believed that antibiotic resistance is a problem." As above, this should have compared public and private sector pharmacists. Which sector had the higher knowledge?
d. In Table 5, Sector was significant for questions 3 and 4. You have an (*) next to the P values, but again state below the Table "Some pharmacists (38.5%) in all sectors significantly believed that having antibiotics under the Food and Drug Act allowed pharmacists to give patients Antibiotics over the Counter in the Private setting and that all antibiotics should be under one Act." Which sector (public or private) had the higher knowledge about these two questions?
e. In Table 6, Sector was significant for questions 1-4, and 6-8. As above please for each whether the public or private sector pharmacists were more knowledgeable?
None
Author Response
- Tables 1-6 show the overall survey results (Yes or No, etc), and also the results of your Chi square (or Fisher exact test) comparing age, gender, experience, and sector (public or private). Any significant P value should be explained. It is not necessary to mark each P value to indicate Chi square or Fisher exact, because you described in Section 4.4 (Data analysis) that the "Fisher’s exact t-test was also used for fields with numbers less than or equal to 5."
- Thank you for this detailed and expert review
- We will leave the indication of Chi 2 vs Fishers
- The reason is to just bring clarity and improve the ease of reading.
- However, the statement about Fisher's test is removed to avoid duplication.
a. In Table 1, Age was significant for questions 2 and 9. You have an (*) next to each P value, and below the Table you can just state "*Younger pharmacists were more knowledgeable than older pharmacists;".
- This is changed
b. In Table 1, Sector was significant for questions 4 and 10. You have a (**) next to each P value, but your caption below the Table states "**Pharmacists in all sectors significantly believed that antimicrobial resistance is not only to antibiotics and that vaccines can prevent unnecessary antibiotic use." This should be a comparison between public-sector and private-sector pharmacists. Which sector was more knowledgeable about question 4, and which was more knowledgeable about question 10?
- This is done in the table and text.
- Table - **Pharmacists in both private and public sectors significantly believed that antimicrobial resistance is not only to antibiotics, but more private sector pharmacists knew that vaccines can prevent unnecessary antibiotic use.
- Text -
When pharmacists’ responses to the statement “antimicrobial resistance results in resistance to antibiotics only” were compared with their sector of employment, a significant association was found for both private and public sectors (p = .017). This was also true for the statement “vaccines can prevent unnecessary antibiotic use and thus antibiotic resistance” and their sector of work, where more private sector pharmacists responded affirmatively (p = .001). This is displayed in Table 1.
c. In Table 4, Sector was significant for question 5. You have an (*) next to the P value, but again state below the Table "*Pharmacists in all sectors and believed that antibiotic resistance is a problem." As above, this should have compared public and private sector pharmacists. Which sector had the higher knowledge?
- The statement about Fishers' exact t-test and the # sign is removed throughout.
-
*Pharmacists, mainly in the private sector, believed that antibiotic resistance is a problem. **Pharmacists mainly in the private sector, agreed that there is abuse of antibiotics.
d. In Table 5, Sector was significant for questions 3 and 4. You have an (*) next to the P values, but again state below the Table "Some pharmacists (38.5%) in all sectors significantly believed that having antibiotics under the Food and Drug Act allowed pharmacists to give patients Antibiotics over the Counter in the Private setting and that all antibiotics should be under one Act." Which sector (public or private) had the higher knowledge about these two questions?
-
*Pharmacists (38.5%) mainly in the private sector, significantly believed that having antibiotics under the Food and Drug Act allowed pharmacists to give patients antibiotics over the counter in the private setting and that all antibiotics should be under one Act.
e. In Table 6, Sector was significant for questions 1-4, and 6-8. As above please for each whether the public or private sector pharmacists were more knowledgeable?
-
* The private sector pharmacists were significantly associated with responses of over-the-counter dispensing and other possibly errant practices, such as dispensing without a prescription or doctors’ advice.
Round 3
Reviewer 3 Report
Thank you. I feel that the results are now much easier for the audience to understand.
None